# Does Car Sharing Contribute to Urban Sustainability from User-Motivation Perspectives?

**Inese Mavlutova** [1,*] , **Jekaterina Kuzmina** [1] , **Inga Uvarova** [2] , **Dzintra Atstaja** [2] , **Kristaps Lesinskis** [2] , **Elina Mikelsone** [2] **and Janis Brizga** [2,3]

1. Department of Economics and Finance, BA School of Business and Finance, LV1013 Riga, Latvia; jekaterina.kuzmina@ba.lv
2. Department of Management, BA School of Business and Finance, LV1013 Riga, Latvia; inga.uvarova@gmail.com (I.U.); dzintra.atstaja@ba.lv (D.A.); kristaps.lesinskis@ba.lv (K.L.); elina.mikelsone@ba.lv (E.M.); janis.brizga@lu.lv (J.B.)
3. Department of Environmental Governance, University of Latvia, LV1586 Riga, Latvia
* Correspondence: inese.mavlutova@ba.lv; Tel.: +371-2952-4701

**Abstract:** Mobility, its current state and development perspectives in the future creates challenges with respect to sustainability, the first of which is the uncontrolled increase in greenhouse gas emissions in the last few decades, while road transport is one of the "sinners" creating long-term negative impact. The second is the dominance of car travel and car usage in the passenger transportation segment before the latest COVID-19 pandemic accelerated environmental problems. Although recent trends show new, greener patterns in consumption, there is still a relatively low share of consumers acknowledging the importance of sustainable and green preferences. This research study aims to investigate car sharing from users' perspectives and to determine the most significant factors influencing their choice of sharing services to ensure upscaling of car sharing and, thus, contribute to urban sustainability. This research study contributes to the overall scientific discussion on car sharing and its role within urban sustainability, particularly with the following: (1) deeper investigation of car sharing and its users motivation perspectives in Latvia; (2) analyses of the most significant motivational factors for car-sharing users and aspects of sustainability; and (3) the insight into the generational differences triggering a number of car-sharing users. The existing and potential users of car sharing were surveyed in order to determine the motivational factors for its usage and attitudes towards it. Socio-demographic variables in statistical analysis were used to identify economic and environmental factors that meaningfully influence the choice of car-sharing services. The results of this study can support further development in new car-sharing business models and the value proposition for consumers in Latvia, as well as preparing policy recommendations on the promotion of sustainable transport. These findings are also useful to academics for the investigation of recent trends in car sharing during the COVID-19 pandemic.

**Keywords:** car sharing; urban sustainability; sustainable transportation; user's motivation; COVID-19

## 1. Introduction

Over the years, decision makers, politicians and active representatives of society have been trying to solve several sustainability problems. Several challenges requiring urgent solutions are particularly related to the sustainable development of the economy, climate change and its consequences to the environmental issues and, last but not least, to the overcrowding of urban space.

International organizations are highlighting the importance of sustainable urban development and mobility as an important element in the context of sustainability. Back in 2017, the United Nations published the *New Urban Agenda*, advocating the necessity for leveraging urbanization for structural transformation, sustainable economic growth, value-added activities and resource efficiency [1].

Similar issues are on the discussion agenda of the European Union (EU). The importance of the sustainable urban transition is addressed, with the Urban Agenda for the EU [2] defining the strategic direction for policy makers, urban development planners and other practitioners ensuring safe, clean, resilient and sustainable urban development.

It is worth considering that sustainable transformation processes cannot be discussed without transportation and mobility issues. According to the documents published by the European Environment Agency [3–5] between 2018 and 2020, the transportation sector contributes to several environmental problems such as noise, climate change, air pollution and other problems related to the negative impact on the quality of life and health of society.

In recent years, more and more citizens are being attracted to modern technologies and other facilities offering better access to services. Moreover, a reasonable portion of consumers is increasingly thinking about their ecological footprint and the impact it has on themselves and those around them. As an alternative, instead of the ownership of goods, individuals, especially of younger generations, prefer networking in order to pool or share the resources, assets or products [6]. Consequently, the sharing economy, including the car-sharing industry, has gained rapid popularity. The sharing economy is strongly related to concepts such as joint consumption [7], collaborative consumption, collaborative endeavours, collaborative economy [8], peer-to-peer economy or business models [9]. Bellotti et al. [9] highlight important links of the sharing economy with the social exchange theory [10,11] that grounds essential roots for the further investigation of motivation and behavior of car-sharing users.

It can be agreed with Bardhi and Eckhardt [6] that the physical ownership of goods is now less desired and does not provide superior respect or recognition among peers. This role is being taken over by the higher respect towards social capital with wider networking, collaboration and resource pooling opportunities, completely changing the social context, values, trust and attitudes of individuals towards their peers.

The last decade has observed the emergence of several various new business models related to shared mobility services, car or scooter-sharing models and bicycle-sharing schemes, mostly in urban areas. In the development of shared business models, technologies are becoming more and more important on the one hand, for instance, with the development of electric and self-driving cars and, on the other hand, the cooperation of individuals within an entire ecosystem, where users and consumers co-create new services and co-deliver the value to peers [6,12].

Existing new technologies and multi-sided platforms allow real-time vehicle tracking, and users or consumers can interact when ordering and receiving services. This creates new patterns of customer behaviour and motivation systems, forcing individuals and organisations to interact, collaborate and trust unknown persons and, even more, technologies and artificial intelligence. As emphasized by Bardhi and Eckhardt [6], access-based consumption or sharing significantly expands the range of potential and existing customers, enabling new consumers who have not previously had the opportunity to purchase certain products or own assets. It also requires the academic society and businesses to study new patterns of customer behaviour in more detail in order to create new value propositions following new trends and customer needs.

Moving from one location to another over short distances, especially within the city, without using a private car and, in doing so, avoiding additional unnecessary costs for parking and other costs when not using the car in the city motivates car users to change their mobility habits and shift to shared mobility services. This allows car drivers to maintain personal intimacy during mobility, as opposed to public transport, but at the same time interacting indirectly with other peers in the digital and physical car-sharing network [9,13].

Car sharing and shared mobility services are topics that have already gained particular interest among researchers, and there is extensive literature available on this topic, investigated by Bruno et al. [14–18]. While most of the academic studies focus on car sharing from perspectives such as transport, environment, ecology, climate impact, the circular

economy and new business models, less research is conducted on the societal behaviour and motivation to use car-sharing services. Furthermore, there is a lack of scientific discourse on car-sharing issues and user motivation in Latvia, but it is substantially important, considering that car-sharing is in its infant stage.

Car sharing ensures shared mobility services which can be used as easily as a private car without owning it. Car sharing has been created as a new business model exploiting the opportunities of digital platforms and smart applications to provide convenient mobility services. Researchers emphasize that car-sharing is one of the most widely used directions in the sharing economy applying different forms of sharing and involving various stakeholders [19].

Important events such as the COVID-19 pandemic changed people's perceptions of transport systems and their use. The pandemic forced people to evaluate and change their everyday rhythm and activities, including their daily movements, over a long period of time and general attitudes towards travel and the use of transport services [20,21].

As the coronavirus pandemic has affected work, leisure, travel, shopping and people's attitudes towards car sharing, it presents both opportunities and challenges for alternative solutions [22,23]. Shared mobility services provide new mobility opportunities while keeping a private space in commonly used vehicles [24], and this issue has been particularly important for consumers during the COVID-19 pandemic.

Regardless of the positive aspects of car-sharing and huge interest on the part of the environmentally aware population, it is still a niche. In Latvia, the phenomenon is less studied, and only some the authors are covering the topic to a limited extent [25–28]. Car sharing and environmentally friendly mobility in Latvia are relatively new concepts [26], and it is something that drivers use relatively rarely. However, this sector has growth potential as the number of shared mobility service providers has been increasing over the last 3 years. It is still unclear what factors motivate users and makes them choose to share a car and use shared mobility services [29].

Osikominu and Bocken [30] describe a specific segment of consumers that prefer a simple lifestyle and mainly live in Western European countries with high incomes and have a good education. They feel more confident in following pro-environmental values and recognise the importance of the positive impacts of pro-environmental attitudes and behaviour on their consumption patterns. Moreover, other researchers proved the direct relationship between education level and environmental concerns. Consumers with a higher education level possess higher tolerance towards environmental concerns, which means that they acknowledge the value of sustainable purchase preferences. These consumers can afford to pay more for "greener" preferences and green purchase decisions prevail [31].

The consumption patterns and preferences in Eastern Europe and especially in Latvia are significantly different from Western European countries due to several reasons. First, the overall income level is on average lower there, thus bringing economic factors to the forefront of their purchase decision making [32]. Another reason is that these countries, including Latvia, have been going through a prolonged period of socialist economic systems where private property is rare and the means of production are of social ownership. This post-socialism experience is deeply rooted and leaves a significant impact on attitudes towards shared services, sharing and co-ownership practices. Over time, these disparities narrow as incomes increase and generations change, especially with the arrival of Generation Z [33]. Drapela [32], in the context of Eastern European countries, has investigated that car-sharing preferences are more visible among the educated urban population, while individuals living in the countryside prefer car ownership. The main triggers encouraging new users of car-sharing services in countries such as Latvia are mainly related to the availability of technological innovations and increase in the use of smartphones with wider possibilities of "one-tap" services [32].

The increase in environmentally positive effects within the transportation system requires a complex approach and promotion of new technologies and more responsible

attitudes from the transportation industry or policy makers and from the citizens and society in general [27]. It is also important to examine the effectiveness of government policies (support measures) for sustainable transportation [34–36]. A much deeper understanding of the motivation factors of car-sharing users and their attitudes towards sustainable transportation would provide a better understanding both for the industry in developing appropriate value propositions for car-sharing users and for policy makers in designing further policies and support instruments.

The lack of such in-depth analysis of car-sharing user attitudes in Latvia has been identified as an important gap in previous research [29]. This study is important for policy makers considering the development of new support policies for the promotion of electric shared mobility services in the coming years. This paper contributes to this discussion by further studying the motivational factors and aspects stimulating the upscaling of mobility sharing services in Latvia.

## 2. Literature Analysis

### 2.1. Role of Car Sharing in Urban Sustainability

The United Nations very actively advocates the necessity of urban sustainability. The new paradigm of urban sustainability encourages not only general sustainable transformation but also seriously raises the issue of citizen involvement, attitudes and responsibility towards a better and more sustainable future of cities, promoting the pro-environmental behaviour of individuals as an extremely important element [1]. The role of environmentally friendly transportation for urban sustainability has also been recognized as an important issue by academic research.

Creating a sustainable urban future requires a partial reduction in the use of conventional transport, especially cars, as well as environmental pressure on society, which is essential for the transition towards a sustainable urban future [37–39]. When talking about the transport system, the quality of the environment and the possibilities for improving it are usually analyzed. The sharing economy is observed as a way to potentially reduce the environmental impact and costs of using products and to increase the availability of transport vehicles [20,40–42].

Mobility sharing makes it possible to meet the mobility needs of citizens and the supply of goods while ensuring positive impacts on the environment and promoting sustainable urban development [27].

The practice of collaborative consumption has been known worldwide for more than thirty years, however, the sharing economy has only recently become a new form of consumption pattern [40,43,44]. The basic concept of the sharing economy is based on an emphasis on the ability, possibility and personal preference of citizens to borrow goods or use services rather than to buy and own them [45,46]. Thus, the sharing economy allows citizens to load under-utilized resources.

Collaborative consumption and sharing have ushered in an unprecedented breakthrough in the field of transport. The joint, autonomous and other forms of mobility impact the way people move.

Car sharing refers to an area of the sharing economy that allows individuals to access and use cars when they need them without purchasing a car.

The literature review of the SCOPUS database regarding car sharing proved that this field has persisted as topical research since 1978. There are 519 scientific documents found by searching with keywords "car sharing". This topic becomes even more topical after 2015 when the number of papers grew rapidly. The content analyses of the literature found in the SCOPUS database revealed that the main authors according to the number of the documents are Bruno et al. [14–18].

As the literature review underlines, car sharing should first of all be considered as a potential opportunity for a more sustainable means of mobility and an opportunity to reduce $CO_2$ emissions, since a reduction in the number of car units allows more efficient utilization [25,47–49]. Dong et al. [50] believe that the development of car sharing not

only satisfies people's diverse travel demands but also brings forward a new solution to facilitate urban sustainability. Researchers explored the advantages of car sharing relative to travellers compared with taxis and concluded that increasing the acceptance of car-sharing will increase the competition between car sharing and traditional taxis [50].

The availability of sharing services also affects traffic safety. This is related to the daily choices of private households, as well as travel, seasonality and selected travel activities. Transport safety is important as it affects the potential costs of social care and health both for individuals and accident victims, as well as rescue and support services [51–53]. A valid question arises as to whether car sharing affects the reduction in car ownership or not [41,54]. The digital platforms play an important role in fostering the sharing economy, allowing consumers to search for sharing service providers more quickly and conduct their business using standard contracts that are already being executed by sharing via online platforms [55].

The importance of networking and various multi-sided platforms was emphasized in the development of car sharing where cooperation between different parties is required, including the involvement of users and consumers in the cocreation of new shared business models, technological facilities and shared services [56].

By analysing the geographical elements of the research papers, we conclude that this is a well-researched topic, especially in developed countries and countries with a high-density population. It is widely known that car sharing is used in the United States, Japan, China, Australia and Korea, and scientists estimate that the number of car-sharing users will increase by 12 million people worldwide in the nearest future [44]. In recent years car sharing has been a growing feature of shared services in Europe, where it is widely observed as contributing to environmental sustainability efforts [40,57].

The academic community investigated various aspects of car sharing as part of the urban transportation system and its efficiency. Boldrini et al. [16] emphasized the characteristics of car-sharing systems, such as how customers use the service over time, which has largely not been explored in the body of research. Researchers analysed one month of online car-sharing map data from a large station-based car-sharing operator in France and proposed a classifier that exploits simple average statistics in order to understand whether the station is profitable or not for the operator [16].

Cocca et al. [15] designed the free-floating car-sharing system based on electric vehicles and contributed to this research direction by finding the optimal placement of charging stations and the design of smart car return policies.

Clemente et al. [17] proposed a user-based solution for vehicle relocation in car-sharing systems and considered different operative conditions that are modelled in a Timed Petri Net framework.

Luo et al. [18] studied an online scheduling problem of using car-sharing applications for trips between an airport and hotels.

Car sharing is also considered and discussed within the context of new business models that require collaboration and changes of the value proposition from the product offered to product-service systems [58]. The European Commission considers that the sharing economy with its collaborative character covers a variety of business models [59]. Moreover, car-sharing business models vary with different collaboration forms such as business to business (also known as B2B), business to consumer (also known as B2C), peer to peer (known as P2P) or other hybrid forms. Irrespective of the form of cooperation and the parties involved, the sharing of assets is the key characteristic [60]. Moreover, Boons and Bocken state that "sharing is the predominant way of organizing the provision for human needs", but it can be organised through nine different sharing forms depending on the revenue or compensation model and stakeholders involved in the collaborative consumption of shared services [58] p. 5.

The COVID-19 pandemic provided a new wave of sharing mobility development: demand for micro-mobility services is growing, both for short distances as well as for low-speed, short-term and on-demand travel. If bicycle rental and bike-sharing were

previously popular, then today both station-based and non-steered or free-floating vehicles, such as shared scooters, are available; the latter includes both stationary electric scooter sharing and moped-type scooter sharing [37,43,61,62].

The content analysis of the keywords demonstrates the frequency of terms used in scientific articles related to car sharing (Figure 1).

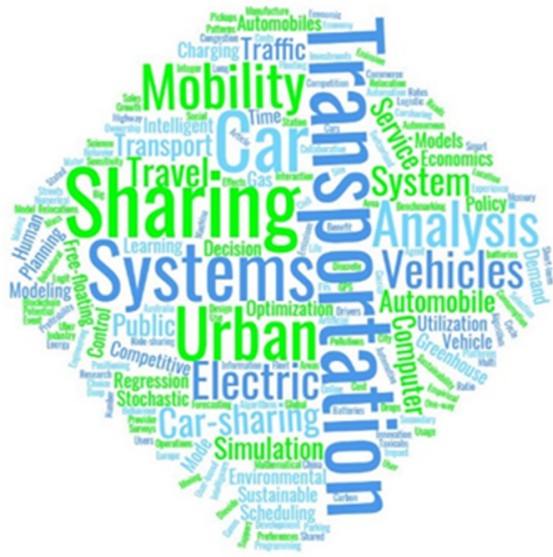

**Figure 1.** Wordcloud based on the SCOPUS articles about car sharing.

The "car sharing system" has been studied in 84 articles, which proves that the body of research relative to existing and new systems is very topical. The keyword "vehicles" (60 articles) proves that different elements connected with transportation are also included in this, but a growing trend is to research "electric vehicles" (44) and "electric automobiles" (59 articles). The keyword "public transport" (44) shows that a major portion of articles also includes elements from public transportation, which could increase the discussion about parities and disparities between car sharing and public transportation. The keywords "optimization" (39) and "traffic congestion" (38) are also well-researched elements. The keyword "the sharing economy" (34 articles) is one of the most frequently mentioned theoretical concepts used for car sharing, and car-sharing studies are more related to "urban transportation" at the moment (30 articles).

Although an additional literature filter with the keyword "motivation" in the SCOPUS database highlighted that only 32 mention the term "motivation" from 519 articles about car sharing, there are almost no studies on the impact of sharing on urban sustainability from the perspective of user's motivation.

In the course of the research study, we concluded that currently one of the topical solutions that would ensure sustainable development of the transportation system is the so-called 3-revolutions or 3-transitions [63]. The three components of the sustainable transition should include the replacement of fuel-powered cars by electric cars; the development and adaptation of automated vehicles (the first two components are based on the necessity of technological innovations); and the move from private ownership towards shared mobility in the transportation system. The third component in the system relates to the behavioural aspect of the users. According to the body of research, the combination of the components of the three transitions could contribute to achieving the goal of sustainability [64–68].

This article has a particular focus on the third component of the system—the transfer from private possession of the vehicle towards shared vehicle or car sharing. Car sharing can be defined as a system that allows renting available cars at any time and for any period so that it provides a substitute to private car ownership by allowing temporary usage on an on-demand basis [69].

Car sharing satisfies the transportation demands of individuals in a sustainable manner, providing lower demand for vehicles and simultaneously reducing traffic and congestion by lowering the number of emissions. Car sharing aggregates the social interconnection and collaboration amongst users of the car-sharing services [70].

Valdemars et al. [51] emphasized the importance of responsibly changing the behaviour of vehicle drivers in order to promote eco-driving and sustainable mobility, while also contributing to transport safety, time saving, encouraging sustainable consumption and contributing to socio-economic development. We agree with these researchers that not only eco-driving but also transport security are underestimated factors contributing to sustainability.

The outbreak of the COVID-19 pandemic has not prevented researchers from discussing the issue of sustainability, but it has encouraged many to consider the research question within the new reality. The latest findings cannot be ignored by the authors of the current research paper and some of the examples are mentioned below. Nevertheless, some authors are searching for insights from early COVID-19 responses with respect to promoting sustainable action [71]. The critical problem discussed is the pandemic's disruptive impacts on social, economic and environmental systems.

Another research area is the assessment of the lockdown impacts on the quality of life in the cities, determined by such factors as air quality, meteorological parameters and mobility data [72]. The most important finding is the proven complexity of the negative environmental effect and the necessity for continuous decarbonization efforts across all emission sectors [73,74].

Moreover, sustainable mobility supported by the COVID-19 pandemic is discussed [75]. The requirement of social distancing has increased the wish to use e-bikes as a means of transport in the city; therefore, there is the growing potential for e-bikes as public transport vehicles in the post-pandemic world [76].

### 2.2. Car Sharing Users—Motivation Factors

In the course of further analysis for the deeper investigation of users' perspective, we added the term "Users" to "car sharing", and 742 articles were selected within the SCOPUS database (TITLE-ABS-KEY ("car sharing") AND (users)). A chronological analysis of keyword frequency and interconnection was performed by using VOSviewer and the results are visualized, indicating clusters of keywords (Figure 2). The chronological application of keywords reveals that during the middle of the last decade, academic discussions on car sharing were dominated by issues related to the management of transport modes and networks, transport infrastructure and safety, accessibility and usability of urban transport systems.

The content analyses of co-occurrence of keywords proves the important interrelation between concepts of car sharing, sustainable transport and sustainable development, and this discussion has become particularly relevant in the last three years. Academics are increasingly discussing research issues of electrical vehicles, positive effects of sustainable transport and the need for appropriate support and policies.

Furthermore, a significant cluster of keywords, which is gradually gaining the interest of academics, is related to the users of car sharing and other sharing mobility services. This research cluster is characterised by the keywords "motivation", "trust", "attitude", "behaviour" and "collaboration", indicating recent research directions.

When exploring car sharing from a user's perspective, we came to the conclusion that there are some studies from the users' point of view, but users' motivation has not been adequately studied in the theoretical body of literature [40,44,61]. For example, Hui, Wang, Sun and Tang built a binominal logit model to analyze the impact of car sharing on the willingness to postpone a car purchase and concluded that car sharing has a positive impact on delaying private car purchasing in China, as it is a country where its citizens have a low level of income [24].

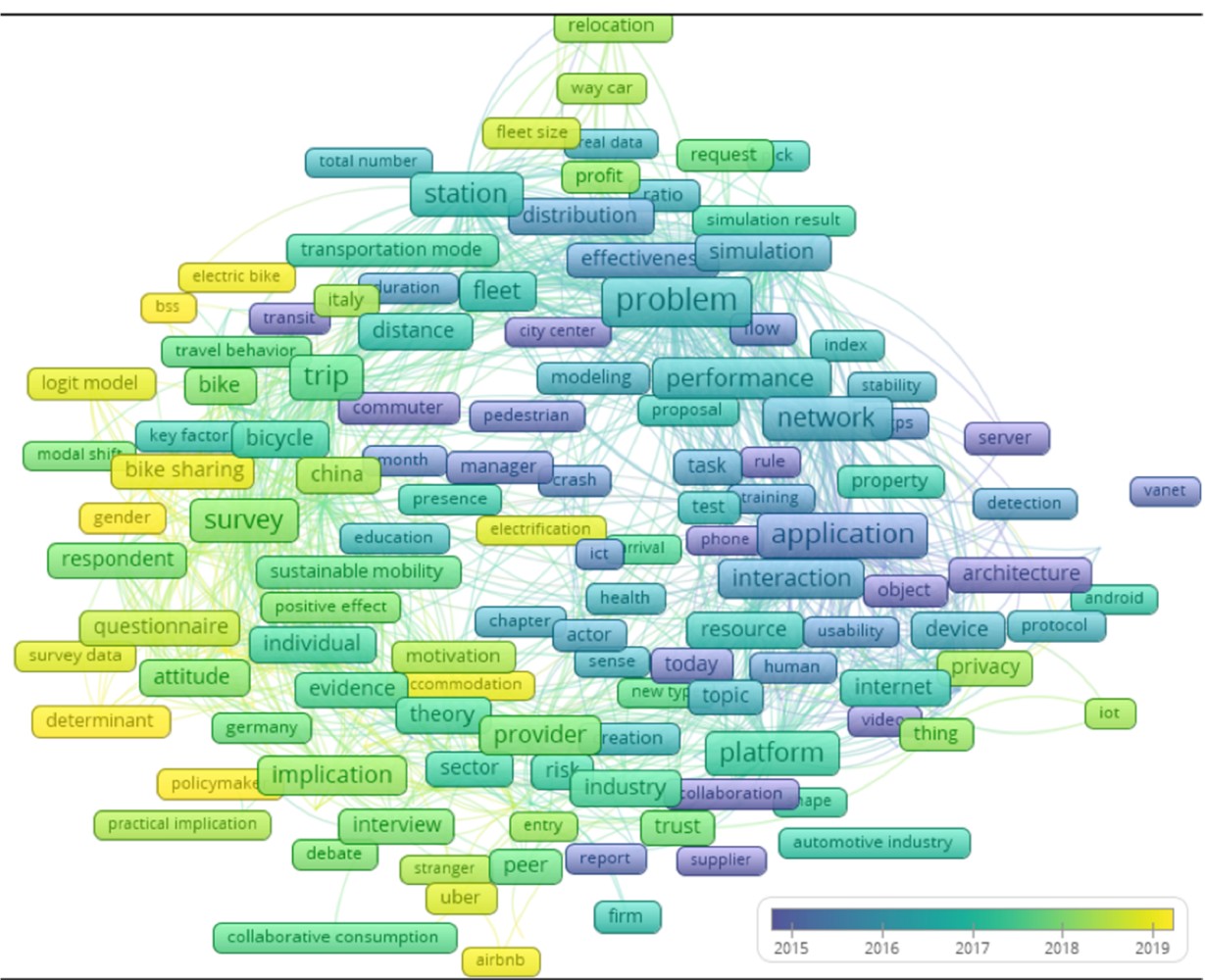

**Figure 2.** The overlay visualization of co-occurrence of keywords "car sharing" and "users" based on the SCOPUS articles using VOSviewer.

Socio-demographic factors, especially "age", are also important for the growth of the car-sharing business models in the last decade. It is obvious that generations Z and Y attach different values to the ownership of tangible goods [60]. As a result, they are increasing the use of shared facilities, and universal access to the property is valued more highly [3,4]. These generations are more open to the voluntary introduction of a more simplistic lifestyle [5] and sustainable consumption patterns, as the purchase and ownership of products or assets are no longer valued so highly in demonstrating affluence.

This article investigates the positive aspects that support the usage of car sharing as discussed in the literature [57,77–79]. Based on the literature analysis, it is possible to state that the motivational factors for car-sharing users could be divided into three groups: first, the environmental benefits, including sustainability aspects [80]; second, the economic benefits, where the issue of cost-saving plays a dominant role; and third, the utility aspect, including the individual perception of users regarding the convenience of services and the level of satisfaction of individual needs.

As the literature analysis reveals, the relevance of the different motivational factors in the perception of the users of a car-sharing service is not equal. Some studies show that rational reasons for participation in car-sharing system are considered as most important: users highlight the significance of saving costs, high personal utility, trust in the service provider and familiarity with it [57,77–80].

This research study will contribute to the existing body of knowledge and research by providing the list of relevant external (or extrinsic, as determined by some authors)

motivational factors and individual (or intrinsic) motivational factors for the usage of car-sharing services. The significance of each factor is determined based on the number of times it is mentioned in existing research; the most important factor occupies a bigger quadrant that is ranged higher.

As Figure 3 shows, the most important factor is the environmental impact that is determined through issues such as a willingness to contribute to the sustainability and positive environmental impact; moreover, the idea of sharing available resources and the initiatives of local politicians could not be disregarded [57,81–85].

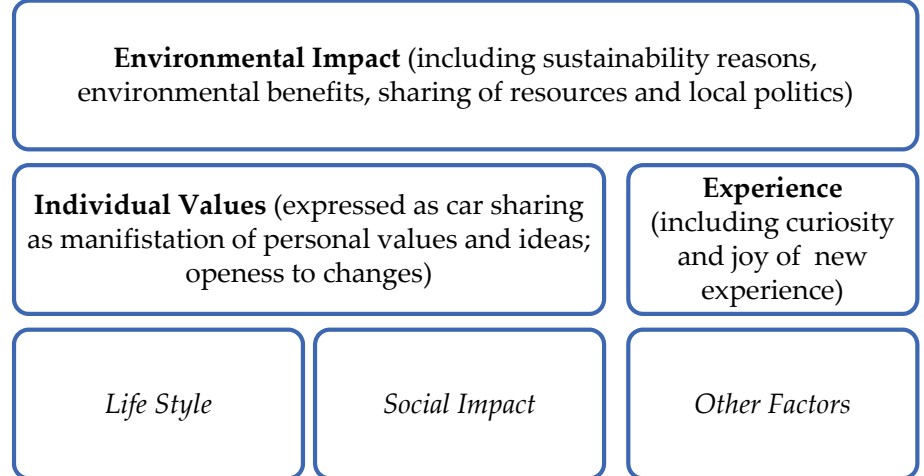

**Figure 3.** Individual motivational factors for utilizing car-sharing services.

Several studies demonstrate that, among adult travellers, the economic benefits are the strongest motivational factors for utilizing car-sharing services, followed by a willingness to be a part of the community by participating in shared consumption [57,81–85]. Furthermore, it was demonstrated that utility dimension factors related to convenience are a prerequisite. According to Figure 4, the main external motivational factors include economic benefits such as cost-saving and time gains and individual utility.

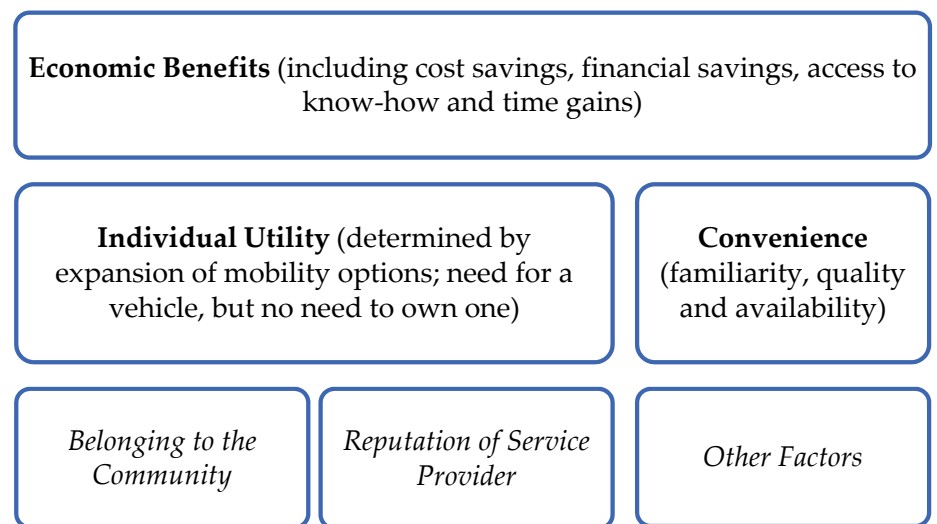

**Figure 4.** External motivational factors for usage of car-sharing services.

The factors determined by the analysis and their relevance are going to be compared with findings from the survey.

## 3. Research Methodology

Car sharing in Latvia is a rather novel mobility service with growth potential. Even though there are different business models in the car-sharing segment across Europe, it is still unclear which factors affect users' behaviour and make them choose car-sharing services. The per capita incomes are lower in Latvia when compared to the EU average and most of the EU countries; thus, this probably brings economic factors to the forefront of decision making. Older generations experienced a prolonged period of a socialist economic system under the Soviet Union until 1991, when private property was rare and means of production belonged to the state. This post-socialism experience is deeply rooted in society and has left significant impacts on attitudes towards shared services, sharing and co-ownership practices. Tambovceva and Titko [29] proved that Latvian people, similar to other Eastern European countries, are reluctant towards sharing services because of the caution about the negligent treatment of someone's property and the lack of trust in unknown persons because of safety and privacy concerns.

The current research study aimed at determining the current situation in the car-sharing segment, as well as providing insights into the behaviour of users and the decision-making process of potential users.

The empirical study was performed in Latvia by using a representative survey of existing and potential users of car-sharing services in order to determine their usage patterns. Latvia is a small country with a population of 1.89 million at the beginning of 2021 [86]. According to the data of the European Automobile Manufacturers' Association (ACEA), the number of passenger cars per 1000 inhabitants (342) in Latvia was the lowest in the EU (569 on average) in 2019 [87]. Considering the potential convergence of this indicator and mobility in general with the EU average, it is important to analyse how an increase in the use of car-sharing services can substitute the necessity for the inflow of new cars in the country and the impact of the COVID-19 pandemic on car-sharing mode choice. The data, based on respondents' attitudes towards car sharing and socio-demographic variables, were used in statistical analysis to identify factors that meaningfully influence the choice of car-sharing services.

The hypotheses of the research are formulated as follows:

- H-1: Car sharing is mostly used by younger and male consumers;
- H-2: Environmental benefits are the most significant factors influencing the choice of car-sharing services and should be considered by shared mobility service providers to ensure upscaling of car sharing;
- H-3: Economic benefits are the most significant factors influencing the choice of car-sharing services and should be considered by shared mobility service providers to ensure upscaling of car sharing in Latvia.

We conducted the research study in the time period from March to May of 2021, with an online questionnaire surveying 364 Latvian urban residents. Data were obtained by asking questions. The original questionnaire consists of three main sections covering the following themes:

- The socio-demographic profile of respondents;
- The experience with the car-sharing services and usage patterns;
- Motivational factors influencing the choice of car-sharing services.

Each section contains various multiple-choice questions, several of which used the Likert scale to quantify the answers. Findings and ideas from other researchers' methodologies and questionnaires related to car-sharing user experience and a range of motivational factors were used to define the survey questions [88]. The results of the survey were interpreted using descriptive statistics. The Chi-squared test method was used to determine whether there is a statistically significant difference between the frequencies in some categories.

Previous studies [85] have acknowledged the lack of research in countries, such as Latvia, with a comparatively lower population density in urban areas. This survey can be

considered as a pilot study to test whether the survey was successfully designed to conduct an in-depth study in the Baltic and other Central Eastern European countries.

The sample focused mainly on the Y and Z generations, but it also included the respondents from generation X and the older population to test hypothesis H-1.

Table 1 shows that 76.9% of respondents represented generations Z and Y, which are believed to be more frequent users of car-sharing services. Gender division is rather close to that of the Latvian population, including 56.9% of female respondents and 42% of male respondents, while 1.1% of surveyed persons decided not to disclose their gender. The majority of surveyed respondents (62.4%) possessed higher education. The majority of all the respondents surveyed (257) are car drivers. It should be noted that 156 or 60.7% of those car drivers have used car-sharing services, but almost all respondents expressed a desire to use shared mobility in the future. Subsequently, responses were obtained only from existing or potential car-sharing users.

**Table 1.** Sociodemographic characteristics of respondents [89].

| Generation | Respondents | Percent (%) |
|---|---|---|
| X and older (42+) | 84 | 23.1 |
| Y (25–41) | 122 | 33.5 |
| Z (18–24) | 158 | 43.4 |
| **Gender** | **Respondents** | **Percent (%)** |
| Male | 153 | 42 |
| Female | 207 | 56.9 |
| Did not want to specify | 158 | 1.1 |
| **Education** | **Respondents** | **Percent (%)** |
| Higher | 227 | 62.4 |
| Secondary general or secondary professional | 137 | 37.6 |

The questions presented to respondents were mostly related to motivation and factors encouraging or impeding the use of car-sharing services. The questions were formulated in order to later be able to divide the answers into four categories of motivational factors—environmental, technological, economical and individual— that are in line with previous research.

## 4. Research Results and Discussion

The results of the survey clearly show that younger people tend to use car-sharing services more often (see Table 2). The difference between the oldest generation researched (X and older) and younger generations (Y and Z) is statistically significant.

As Table 3 indicates, the statistically significant difference by generations was found after conducting a Chi-squared test for these data and obtaining the result "*p*-value < 0.001".

The results of the survey show that the use of car-sharing services is more common among the male population (see Table 4).

As Table 5 indicates, the statistically significant difference by gender was found after conducting a Chi-squared test for these data and obtaining the result "*p*-value < 0.001".

Thus, H-1, stating that car sharing is mostly used by younger and male consumers, turns out to be true.

The respondents were also asked to rate various motivational factors for car-sharing usage on a Likert scale from 1 to 6, where one describes "not an important factor" but six describes a "very important factor". As Table 6 shows, the location of a car (5.51) and price (4.89) turned out to be the most important factors, with ecological footprint scoring just 3.16.

**Table 2.** The proportion of car-sharing service users by generation [89].

| Generation | | Users | Not Users | Total |
|---|---|---|---|---|
| Generation X | respondents | 17 | 67 | 84 |
| | % | 20.2 | 79.8 | 100 |
| Generation Y | respondents | 53 | 69 | 122 |
| | % | 43.4 | 56.6 | 100 |
| Generation Z | respondents | 86 | 72 | 158 |
| | % | 54.4 | 45.6 | 100 |
| Total | respondents | 156 | 208 | 364 |
| | % | 42.9 | 57.1 | 100 |

**Table 3.** Chi-squared test for usage of car-sharing service by generation [89].

| | Value | df | *p* |
|---|---|---|---|
| $X^2$ | 26,207 | 2 | <0.001 |
| N | 364 | | |

**Table 4.** The proportion of car-sharing service users by gender [89].

| Gender | | Users | Not Users | Total |
|---|---|---|---|---|
| Male | respondents | 84 | 69 | 153 |
| | % | 54.9 | 45.1 | 100 |
| Female | respondents | 71 | 136 | 207 |
| | % | 34.3 | 65.7 | 100 |
| Did not want to specify | respondents | 1 | 3 | 4 |
| | % | 25.0 | 75.0 | 100 |
| Total | respondents | 156 | 208 | 364 |
| | % | 42.9 | 57.1 | 100 |

**Table 5.** Chi-squared test for usage of car-sharing service by gender [88].

| | Value | df | *p* |
|---|---|---|---|
| $X^2$ | 15.775 | 2 | <0.001 |
| N | 364 | | |

**Table 6.** Ranking of motivational factors for usage of car-sharing services [89].

| | Mean | Std. Deviation |
|---|---|---|
| Location of a car | 5.513 | 0.891 |
| Price | 4.885 | 1.368 |
| Responsive customer service | 4.462 | 1.526 |
| Comfort | 4.071 | 1.296 |
| Car size | 3.513 | 1.526 |
| Power of a car | 3.212 | 1.455 |
| Car design | 3.192 | 1.451 |
| Ecological footprint | 3.16 | 1.568 |
| Related to sales campaigns | 2.929 | 1.745 |

By analysing the motivating factors of users more thoroughly, these factors were divided into three groups: first, the environmental benefits, including sustainability aspects; second, the economic benefits related to cost saving; third, the economic benefits related to utility. The answers provided by the respondents showed that car-sharing service users are highly motivated by the accessibility of a car (78.8%), price (57.7%), comfort (36.5%) and possible driving distance (36.5%) and other factors are of significantly less importance (see Figure 5).

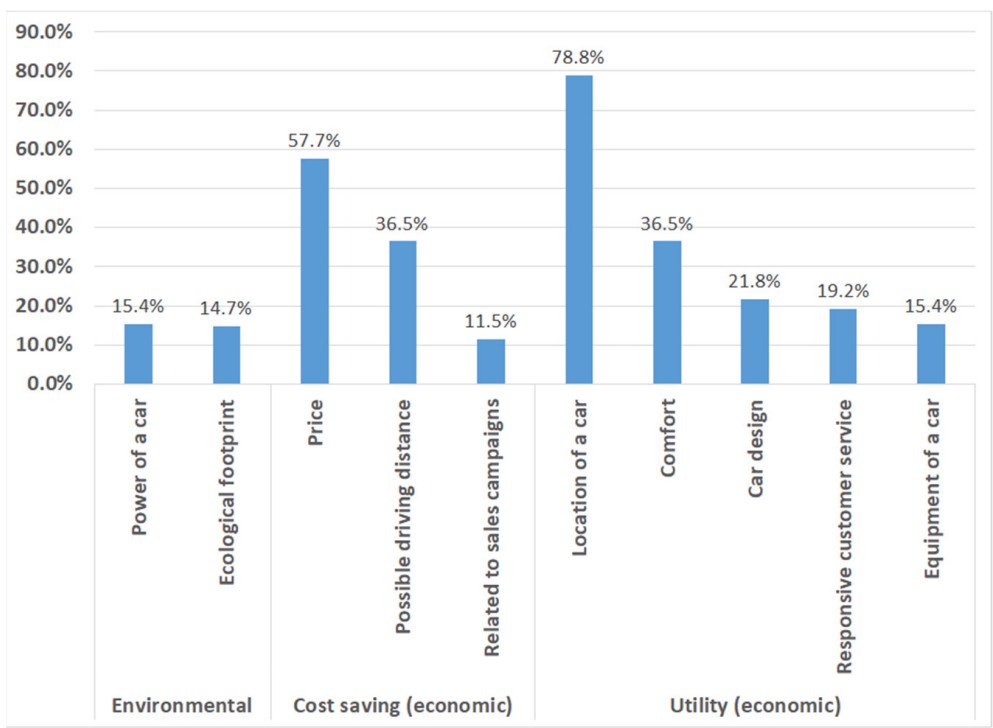

**Figure 5.** Motivational factors for utilizing car-sharing services (several answers possible) [89].

Only 14.7% of respondents chose ecological footprint as a motivating factor to use a car-sharing service. After conducting a Chi-squared test for this factor ($p = 0.527$), no statistically significant differences were found between the generations observed.

We hereby confirm hypothesis H-3: Economic benefits are the most significant factors influencing the choice of car-sharing services, and they should be considered by business providers to ensure upscaling of car sharing in Latvia. Thus, the hypothesis H-2—environmental benefits are the most significant factors influencing the choice of car-sharing services and should be considered by business providers to ensuring upscaling of car sharing in Latvia—turns out not to be true and can be rejected.

In order to identify specific factors that might stimulate the usage of car-sharing services, the authors found out the reasons that deter respondents from using car-sharing services (see Figure 6).

The respondents were asked to rate various factors hindering car-sharing usage once more on a Likert scale from 1 to 6. Two factors stand out: difficulty in finding a car and having no place to park a car. These two factors are heavily related to the infrastructure aspects that are strongly controlled and developed by municipalities and central governments.

After conducting a Chi-squared test for these data, no statistically significant differences were found between generations observed.

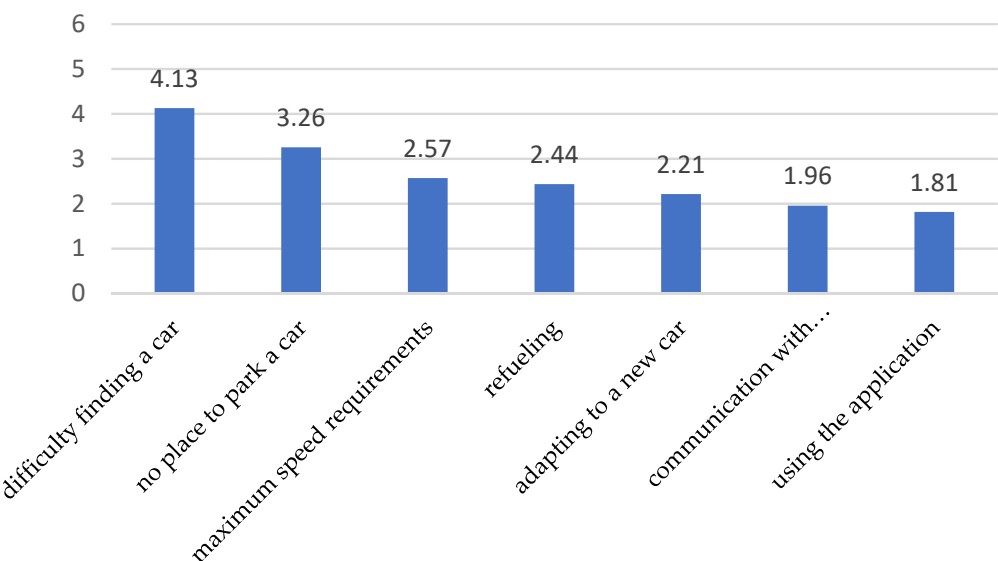

**Figure 6.** Factors hindering the usage of car-sharing services (6—very important; 1—not important) [89].

The authors also tested the impact of the COVID-19 pandemic on respondents' demand for car-sharing services. The results of the survey indicate a decrease in the use of car sharing during the pandemic. The proportion of those who did not use the service during the pandemic rose to 23.1% compared to 13.5% before the pandemic. Moreover, the proportion of everyday users decreased from 6.4% to 3.8% during the pandemic (see Figure 7).

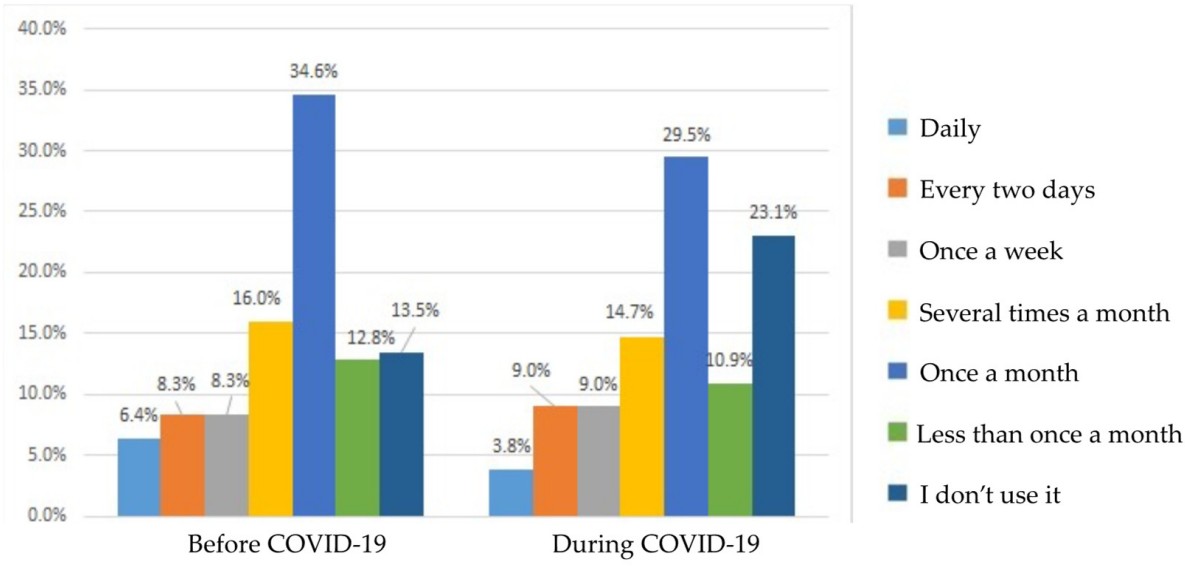

**Figure 7.** The frequency of the usage of car-sharing services before and during the COVID-19 Pandemic [89].

The authors propose the following conceptual model of positive and negative factors influencing the decision to use car-sharing services that could be used as a theoretical background for future studies in other regions (see Figure 8). This conceptual model was developed by assuming previous research [88] and addressing the actual situation in Latvia for the year 2021. By referring to the classification provided by Malichová et.al. [88] with respect to the internal and external influencing factors, the current conceptual model porposes more detailed classification and introduces new groups of factors related to technological and economic issues, the environmental impact and the individual experience.

While in the previous research study [88], most of these factors were considered as internal prihological factors, the development trends of the customer preferences, attitudes and behaviour require more explicit and detailed perception.

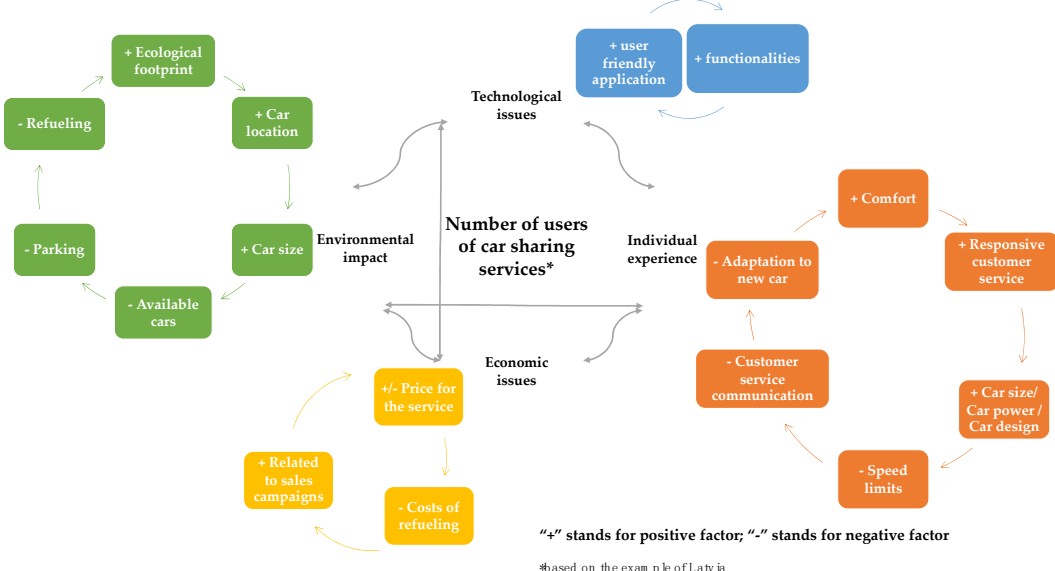

**Figure 8.** The conceptual model of positive and negative factors influencing number of users of car-sharing services.

Each of four factors is dependent on a few subfactors that are marked either with "+" in the case of positive influences on the users' decisions, resulting in the increase in the number of users of car-sharing services, or with "−" in case of negative influences on the users' decisions, resulting in the decrease in the number of users of car-sharing services. The "+" or "−" sign is placed into brackets and does not indicate the positive or negative strength of the particular subfactor. It is worth considering that the subfactors contain only those mentioned in the research study (using questioner) conducted by the authors; therefore, additional subfactors could be considered if necessary. Moreover, one could note that the factors and subfactors are interrelated; the changes in one of the factor/sub-factor are going to affect the others.

The authors agree with Liao et al. that car sharing can be considered as an alternative sustainable mobility solution because it reduces the number of private vehicles and because car-sharing fleet owners are increasingly choosing to buy electric cars. All of these solutions have a positive effect on reducing pollution and $CO_2$ emissions [90].

The study reveals that car sharing is mostly used by younger people; this corresponds to the findings by Prieto et al. [91].

Several studies show that rational reasons for participation in car-sharing systems are considered as the most important: users highlight the significance of cost-saving; high personal utility; and trust in the service provider and familiarity with it [64,65,92].

If the accessibility of car, price of service, comfort and possible distance to drive are rational reasons in a spectrum of motivational factors for users, the survey results are consistent with the previous studies mentioned above.

The findings from this survey comply with previous findings by several researchers [56,67–70] that, among adult travellers, the economic benefits are the strongest motivational factors for utilizing car-sharing services.

However, it should be noted that the relatively low importance given to the solution of ecological problems does not mean that the use of car-sharing practices could not be stimulated by policies that focus on motivating factors. For example, both central and local governments can participate in building appropriate infrastructure, thus improving

accessibility to the service, as well as providing some financial stimulus through tax incentives or other policy tools resulting in lower prices to end users of the service.

As it follows from the research results, investment in infrastructure along with the efforts to provide better accessibility relative to cars by car-sharing service operators, especially those with electric engines, could substantially contribute to sustainable mobility in urban areas.

The requirement of social distancing was a trigger to reconsider the usage of e-bikes as a means of transport in the city: one can note the increased role of e-bikes as a reliable mode of transport and the growing potential for e-bikes as substitutes for public transport in the post-pandemic world [93].

## 5. Conclusions

We determined the most significant factors influencing the choice of car-sharing services from the users' perspective and concluded that car sharing is an environmentally friendly solution promoting urban sustainability. The motivational factors of car-sharing users are usually divided into three groups: environmental benefits, including sustainability aspects; economic benefits, where the cost-saving issue is dominant; and utility aspect, including the individual perception of users regarding the convenience of services and the level of satisfaction. The authors contributed to the study by providing a list of relevant external (or extrinsic, as determined by some authors) motivational factors and individual (or intrinsic) motivational factors for the usage of car-sharing services.

We conducted the study among the Latvian population, and its results are consistent with previous studies that stated that socio-demographic factors are important for the growth of the car-sharing model, and it is obvious that generations Z and Y, especially males, attach different values to the ownership of tangible goods. As a result, they are increasing the use of shared facilities, and universal access to property is valued more highly.

The study confirms the observations in previous studies by different researchers that economic benefits are the most significant factors influencing the choice of car-sharing services. Relatively low importance is given to the solution of environmental problems. The use of car-sharing practices might be stimulated by policies that focus on the identified motivating factors. For example, both central and local governments can participate in building appropriate infrastructure; thus, this would improve accessibility to the service and provides some financial stimulus through tax incentives or other policy tools resulting in lower prices to end users of the service, as these turned out to be the most important factors.

Moreover, the study revealed that the analysis of economic factors has to be performed from the perspective of purely economic factors and utility factors (including individual perception and emotional components). Thus, the relevant factor could be determined more precisely and contribute to a better understanding of the phenomena.

We have developed a framework of positive and negative subfactors influencing the decision on whether to use car-sharing services based on our own research and analysis of data obtained from using questionnaires. The developed construct could be used as the theoretical background for future studies in other regions; therefore, the authors are contributing to the development and discussion of the issue. A deeper understanding of motivational factors will allow service providers to satisfy customer needs better, as well as propose a different kind of shared mobility services, thus resulting in the development and promotion of various sustainable transportation modes.

The COVID-19 pandemic also created particular interest in the research within the concept of smart cities as a sustainable city model, ensuring a change in urban mobility. Innovations based on micromobility solutions (sharing scooters or bicycles) became the most topical issues.

The results of the study have some limitations and further research is recommended. The study was mainly designed to look at social-demographic factors such as age and gender and does not include an essential analysis of the correlation between other factors

influencing demand decisions, e.g., education and income levels. As further research following the results of this study, an analysis of policy instruments to motivate car sharing would be highly desirable.

**Author Contributions:** Conceptualization and methodology, I.M. and I.U.; analysis and investigation, J.K., E.M., D.A. and I.U.; writing—original draft preparation, J.K., I.U. and I.M.; writing—review and editing, J.B., I.U. and D.A.; supervision and project administration, I.M.; the survey and data acquisition, K.L. and I.U.; obtaining funding support, I.M. and D.A. All authors have read and agreed to the published version of the manuscript.

**Funding:** The paper was funded by Latvian Council of Science, the project "The Impact of COVID-19 on Sustainable Consumption Behaviours and Circular Economy" (Nr. lzp-2020/2-0317).

**Institutional Review Board Statement:** Not applicable.

**Informed Consent Statement:** Not applicable.

**Data Availability Statement:** Not applicable.

**Acknowledgments:** The authors thank Arnis Lazdins for the assistance provided with respect to developing the questionnaire and organising the survey.

**Conflicts of Interest:** The funders had no role in the design of the study; in the collection, analyses or interpretation of data; in the writing of the manuscript or in the decision to publish the results.

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
