# Peer review of "Does Car Sharing Contribute to Urban Sustainability from User-Motivation Perspectives?"

_sustainability, doi:10.3390/su131910588_

Round 1

Reviewer 1 Report

The manuscript entitled “Does Car Sharing Contribute to Urban Sustainability from User-Motivation Perspectives? " and the authors focused on the instrumental factors that can impact users willingness to  use car sharing systems.

However, the authors aimed at determining and attitudes towards use of car sharing, but the collected data and analysis method are not supporting the objectives of research.

The authors in the abstract of manuscript claimed that the socio-demographic characteristics of respondents were collected and analyzed, but this information have not been presented in the paper if such data have been already collected. For instance, In this paper only one socio-demographic characteristics “age” has been considered in data analysis and other factors such as gender, occupation, residence location (e.g. urban, suburbs), level of income influencing people’s intention towards car-sharing are missing data analysis section

Despite very interesting objectives of manuscript, but the proposed research questions (3 hypothesis) are not supporting these objectives. Authors should align somehow the proposed research hypothesis with the research objectives.

In line 24 authors indicate that they aim at “deeper investigation of perspective within the countries of Central Eastern Europe” while the research focuses only on Latvian case study. So, it’s better to revise it and make it more clear for readers about the case study.

In line 30, authors claim that results of study for Latvian case study can support further developing car sharing business model and  the  value proposition for consumers of central easter Europe. Considering a sizable variation of value proposition of consumers across EU Member States, then I would suggest to change the CEE countries with Latvia.

The literature section is very long  ( 9 pages) and very hard to read. Some references are out of scope. So this section can become more concise by removing redundant texts (e.g. in lines 184-309.) and adding some relevant reference from European studies in domain of transport supporting research objectives such as

Ramos, É.M.S., Bergstad, C.J., Chicco, A. et al. Mobility styles and car sharing use in Europe: attitudes, behaviours, motives and sustainability. Eur. Transp. Res. Rev. 12, 13 (2020). https://doi.org/10.1186/s12544-020-0402-4

Malichová, E.; Pourhashem, G.; Kováčiková, T.; Hudák, M. Users’ Perception of Value of Travel Time and Value of Ridesharing Impacts on Europeans’ Ridesharing Participation Intention: A Case Study Based on MoTiV European-Wide Mobility and Behavioral Pattern Dataset. Sustainability 2020, 12, 4118. https://doi.org/10.3390/su12104118

In line 318,

  • keep the number of car minimum can be replace by keep use of cars minimum.
  • In transit can be substitute by daily travel.
  • it is not clear what authors mean by social pressure. The social pressure term needs to be clarified.

Regarding the research methodology, data description, authors should clearly explain about the method of data acquisition/question, Types of answers. For instance, impact of COVID-19 pandemic on Car sharing mode choice has been mentioned only in Section 3 (lines 619-623. This factor stands alone without further clarification.

Authors should make a clear explanation about the questions asked regarding the environmental and economic benefits by providing a table of analyzed variables for these two hypotheses.
In this paper we can see only in figure 5 the importance of each factors based on respondence answers. So it would better if authors could make a distinction between the environmental and economic benefit in terms of their significance on car-sharing mode choice using the weighting method along with ranking of motivational factor for each instrumental motivations.  

Figure 7 is not readable. The quality of figure should be improved.

After applying the comments, authors should also further elaborate on Conclusions section based on about the contribution of their research findings for promotion of more sustainable transport modes.

Overall, the manuscript has major flaws that needs a major revision to be accepted.

Reviewer 2 Report

  • The study is based on a survey carried out in Latvia and aims at studying the motivation of users for preferring car sharing against car ownership. Influencing factors as for instance age, positive motivational and hindering factors are analyzed. The statistical analysis is descriptive.

  • Although the analytical level for the paper is modest the paper gives interesting information on the preferences of people and their motivation for choosing car sharing options.

  • Section 2 (literature analysis) is too long and presents much information which is not needed for understanding the methodological approach. Sub-section 2.1 is most heterogenous and presents a patchwork of different models which are not related to the topic. I recommend cancelling this sub-section.

  • The paper appears appropriate to be published in Sustainability subject to the recommended revision. Minor language check is also recommended.

Round 2

Reviewer 1 Report

Authors have addressed my concerns. The manuscript has been considerably improved. I recommend that the paper should be accepted in the present form.